# Definitions and Operationalization of Mental Health Problems, Wellbeing and Participation Constructs in Children with NDD: Distinctions and Clarifications

**DOI:** 10.3390/ijerph18041656

**Published:** 2021-02-09

**Authors:** Mats Granlund, Christine Imms, Gillian King, Anna Karin Andersson, Lilly Augustine, Rob Brooks, Henrik Danielsson, Jennifer Gothilander, Magnus Ivarsson, Lars-Olov Lundqvist, Frida Lygnegård, Lena Almqvist

**Affiliations:** 1CHILD, School of Health and Welfare, Jönköping University, 55110 Jönköping, Sweden; annakarin.andersson@ju.se (A.K.A.); frida.lygnegard@ju.se (F.L.); 2The Swedish Institute for Disability Research, 58183 Linköping, Sweden; lilly.augustine@ju.se (L.A.); henrik.danielsson@liu.se (H.D.); magnus.ivarsson@liu.se (M.I.); lars-olov.lundqvist@regionorebrolan.se (L.-O.L.); 3Department of Paediatrics, The University of Melbourne, Melbourne 3052, Australia; christine.imms@unimelb.edu.au; 4Bloorview Research Institute, Torornto, ON M4G 1R8, Canada; gking27@uwo.ca; 5CHILD, School of Education and Communication, Jönköping University, 55110 Jönköping, Sweden; 6School of Clinical and Applied Sciences, Leeds Beckett University, Leeds LS1 3HE, UK; r.b.brooks@leedsbeckett.ac.uk; 7Department of Behavioural Sciences and Learning, Linköping University, 58183 Linköping, Sweden; 8School of Health, Care and Social Welfare, Mälardalen University, 72123 Vasteras, Sweden; jennifer.gothilander@mdh.se (J.G.); lena.almqvist@ju.se (L.A.); 9University Health Care Research Center, Faculty of Medicine and Health, Örebro University, 70185 Örebro, Sweden

**Keywords:** concept, mental health problems, mental health, wellbeing, participation, concept

## Abstract

Children with impairments are known to experience more restricted participation than other children. It also appears that low levels of participation are related to a higher prevalence of mental health problems in children with neurodevelopmental disorders (NDD). The purpose of this conceptual paper is to describe and define the constructs mental health problems, mental health, and participation to ensure that future research investigating participation as a means to mental health in children and adolescents with NDD is founded on conceptual clarity. We first discuss the difference between two aspects of *mental health problems*, namely mental disorder and mental illness. This discussion serves to highlight three areas of conceptual difficulty and their consequences for understanding the mental health of children with NDD that we then consider in the article: (1) how to define mental health problems, (2) how to define and assess mental health problems and mental health, i.e., wellbeing as separate constructs, and (3) how to describe the relationship between participation and wellbeing. We then discuss the implications of our propositions for measurement and the use of participation interventions as a means to enhance mental health (defined as wellbeing). Conclusions: Mental disorders include both diagnoses related to impairments in the developmental period, i.e., NDD and diagnoses related to mental illness. These two types of mental disorders must be separated. Children with NDD, just like other people, may exhibit aspects of both *mental health problems* and wellbeing simultaneously. Measures of wellbeing defined as a continuum from flourishing to languishing for children with NDD need to be designed and evaluated. Wellbeing can lead to further participation and act to protect from mental health problems.

## 1. Introduction

Children with impairments are known to experience more restricted participation than other children [1]. It also appears that low levels of participation are related to a higher prevalence of mental health problems in children with neurodevelopmental disorders (NDD) [2,3]. NDD is a group of early onset conditions associated primarily with the functioning of the neurological system and brain, including diagnoses such as attention-deficit/hyperactivity disorder, autism, and intellectual disability [4]; sometimes, cerebral palsy is also seen as an example of NDD, although it is primarily presented as a motor disorder in DSM-V. NDDs lead to impairments in physical, social, or academic functioning, which affect different aspects of participation.

In the Family of Participation Related Constructs (fPRC) framework, participation is described as consisting of two dimensions: physical or virtual attendance in activities, which is seen as a necessary prerequisite for the second dimension, involvement while attending the activity [5]. The fPRC framework builds on the International Classification of Functioning, Disability, and Health (ICF) definition of participation [6] by specifying two separate dimensions of attendance and involvement. It has been suggested that participation is a determinant of mental health [7]; however there is not currently a deep understanding of the relationships between mental health and participation within NDD.

While a higher prevalence of mental health problems is reported for children with impairments [3,8], especially for children with NDD, the suppositions behind the higher prevalence are implicit rather than explicit. In medical literature, mental health is commonly defined as the absence of mental health problems [9], but without a clear definition of the construct of mental health being provided. Based on Jahoda [10], Westerhof and Keyes [11] suggest that mental health is a positive phenomenon that is more than the absence of mental health problems. They define mental health in terms of hedonistic and eudaimonic wellbeing, which is also the definition we will defend in this article.

Children and adolescents with NDD seldom receive non-pharmacological mental health interventions specifically aimed at reducing mental health problems [12], although studies aimed at increasing subjective wellbeing with the help of mindfulness intervention exist [13]. We suggest that participation interventions—that is, those that aim to improve attendance or involvement in varied life situations—can be implemented to strengthen mental health as well as indirectly prevent or decrease mental health problems in children and adolescents with NDD. Thus, interventions aimed at increasing participation may be a means to increase perceived mental health [14,15]. To test this proposal, the conceptual relations between mental health problems, mental health (defined as wellbeing), and participation need to be clarified. The purpose of this paper is to describe and define these constructs to ensure that future research investigating participation as a means to mental health is founded on conceptual clarity. 

To achieve our purpose, we first discuss the difference between two aspects of mental health problems, namely mental disorder and mental illness. This discussion serves to highlight three areas of conceptual difficulty and their consequences for understanding the mental health of children with NDD that we then consider in the remainder of this article: (1) how to define mental health problems (and delimit them from mental disorders and mental illness), (2) how to define and assess mental health problems and mental health (i.e., wellbeing) as separate constructs, and (3) how to describe the relationship between participation and wellbeing. We then discuss the implications of our propositions for measurement and the use of participation interventions as a means to enhance mental health (defined as wellbeing), thus proposing a way forward.

## 2. Issues of Classification of Mental Disorders and NDD in Diagnostic Manuals

In the ICF, aspects of functioning disability and health are classified as body structure and function, activity, and participation, thereby building on a bio-psycho-social model. The ICF is supposed to be a supplement to the diagnostic manuals used in medicine disorders, the International Classification of Diseases (ICD-11) [4], and mental disorders, the Diagnostic Systems Manual (DSM 5) [16]. These diagnostic systems include NDD, for example, intellectual disability and ADHD, within the classification of types of mental disorder, along with schizophrenia, depression, and disorders due to substance abuse [6]. In the ICD-11, mental disorders are defined and described in chapter 6: Mental, behavioral, or neurodevelopmental disorders. This chapter states: 

“*Mental, behavioral and neurodevelopmental disorders are syndromes characterized by clinically significant disturbance in an individual’s cognition, emotional regulation, or behavior that reflects a dysfunction in the psychological, biological, or developmental processes that underlie mental and behavioral functioning. These disturbances are usually associated with distress or impairment in personal, family, social, educational, occupational, or other important areas of functioning.*”(Chapter 6, p.1 ICD-11, 2020)

In this description, the relationship between mental disorders and everyday functioning is emphasized. The definition provides core aspects to look for when diagnosing mental disorders (cognitive, emotional, or social abilities and behavior), but it is only in the subclassifications that a distinction is made between NDD and mental illnesses such as depression or general anxiety disorder. In discriminating between different mental disorders, the ICD-11 states that NDD is characterized by symptoms that emerge in the developmental period; however, this characteristic is not unique to NDD, as other mental disorders may also present in the developmental period. 

Because classification systems like ICD-11 and DSM-V are designed to define “disease” or “condition”, they do not define positive mental health and do not explicitly make a distinction between bio-psycho-social levels. Unless outcomes in terms of mental health are clearly defined, it is difficult to assess mental health other than as the absence of a disease or condition. Unless mental health outcomes are clearly defined, it is difficult to plan interventions aimed at improving mental health for children and adolescents with NDD, because no positive outcome other than lack of mental health problems is described. In Table 1, definitions of the terms used in this paper are presented.

## 3. Core Difficulties with the Definition and Operationalization of the Constructs Defined for Children with NDD

We identify three core difficulties with definitions and operationalization of these constructs: 

*Problem 1*. How to define mental health problems in children with NDD and delimit them from mental disorders and mental illness. 

*Problem 2.* How to define and assess mental health problems and wellbeing as separate constructs in children with NDD. 

*Problem 3.* How to describe the relationship between participation and wellbeing in children with NDD. 

### 3.1. Problem 1: Distinguishing Mental Health Problems from Mental Illness and Mental Disorders in Children with NDD

We propose that any construct and measure of mental health problems should be equally applicable for children and adolescents, regardless of cognitive and physical impairment, and without defining mental illness as equal to the impairment

The constructs mental disorders, mental illness, and mental health problems all concern problems related to mental function. In this section, we first discuss the relationship between mental disorders and mental illness and thereafter the relationship between mental illness and mental health problems.

#### 3.1.1. Mental Disorders and Mental Illness—The Example of NDD 

The World Health Organization [6] provides a general definition of a health condition as: “an umbrella term for disease (acute or chronic), disorder, injury or trauma. A health condition may also include other circumstances such as pregnancy, ageing, stress, congenital anomaly, or genetic predisposition” [6], (p. 228). The definition of a mental disorder seems to build on the general definition of a health condition [4]. Mental illness is not formally defined in the ICD-11 but is, in everyday language, used to describe mental disorders other than NDD, such as mood disorders, anxiety, and fears [9]. In the ICD-11, NDD diagnoses are included as examples of mental disorders. The definition of NDD (see Table 1) stresses that NDD concerns cognitive and behavioral problems that arise during the developmental period. This implies that NDD has qualities distinct from other mental disorders and is, therefore, a sub-category of its own. One could argue, however, that there is a good case for not separating different diagnoses based on whether they indicate cognitive difficulties or not, since, for example, cognitive impairments are also a part of the clinical picture in severe depression (in that case time-limited), schizophrenia (more permanent in nature), and addiction. The fact that NDD primarily concern intellectual, motor, and/or social functions that are more or less permanent, compared to mental health problems, and arise during the developmental period provides an argument for separating NDD from mental disorders that can be described as mental illness. 

#### 3.1.2. Mental Illness and Mental Health Problems

In their conceptual analysis of mental health, mental disorders, and mental health problems, Bremberg and Dalman [9] illustrate the overlap between the constructs using a figure. We have adapted their figure by making a distinction between mental disorders and mental illness to illustrate our argument (made above) that NDD does not necessarily involve mental illness (see Figure 1).

As Figure 1 illustrates, in many cases mental health problems overlap with wellbeing: mental health problems are a normal part of people’s lives, but so is wellbeing. However, mental health problems also partly overlap with mental illness: having persistent mental health problems in childhood increases the probability of being diagnosed with a mental illness in adulthood [21]. Figure 1 also shows that mental illness is completely subsumed within mental health problems, but some mental disorders (e.g., NDD) do not automatically overlap with mental illness or mental health problems. 

#### 3.1.3. Difficulties in Defining and Operationalizing Mental Health Problems Following from Conceptual Diffuseness 

When definitions of constructs or diagnoses, such as mental health and NDD, are restricted to separate and different levels of the bio-psycho-social model, there is no risk for confusion or overlap, e.g., between traumatic brain injury, which is defined primarily on basis of etiology on the biological level, and behavior problems (as measured by CBCL). However, the risk for confusion between symptom-based diagnoses, such as NDDs, and mental health constructs operating at the same level(s) of the model, is more probable.

An example of a very practical consequence of the conceptual overlap between a mental disorder and a mental health problem, which may create conceptual confusion, is how authors define mental health problems when screening children with NDD. A study by Bailey et al. [22] used two indexes of mental health difficulties, as suggested for typical populations [23], namely internalizing (emotional and peer problems) and externalizing (conduct and hyperactivity) mental health difficulties, based on the definition of a mental disorder. The same type of indices is used with other “problem behavior” screening instruments, such as the Child Behavior Checklist (CBCL) [24]. Using this operationalization, children with intellectual disability have significantly more mental health problems than typically functioning children. However, this fusion of what may be factors related to cognitive impairments (i.e., communication/peer problems and hyperactivity) and behavior problems (i.e., emotion and conduct problems) may in fact lead to an overestimation of the prevalence of externalizing and internalizing mental health problems among children and adolescents with diagnoses of NDDs.

Longitudinal studies of behavior problems involving children and adolescents with diagnosed NDD [21] and children with self-reported NDD problems [25] suggest that there is a continuum of chronicity for common mental disorders. These studies suggest that problems not necessarily related to mental illness but to consequences of cognitive impairments (hyperactivity and peer problems) have stronger stability over time than mental disorders that can be described as mental illness (i.e., anxiety and depression). 

Mental illness is seen as a severe and intensive type of mental health problem, situated completely within the broader circle of mental health problems (see Figure 1). The point at which mental health problems are severe enough to be diagnosed as a mental illness is debatable and somewhat arbitrary [9]. Most mental health problems have a shorter duration and less severity than a mental illness. Mental illnesses are primarily identified through diagnostic interviews where the person is required to meet certain criteria regarding the severity and persistence of problems to receive a diagnosis.

Longitudinal studies of mental health problems are needed to investigate relationships between mental health problems, such as conduct problems, anxiety, and sadness/depression (mental illness), and NDD, a mental disorder separate from mental illness.

#### 3.1.4. Mental Health Problems and Wellbeing (Mental Health) over the Life Course 

Mental health problems vary over the life course with certain periods, such as adolescence, having both biological change and changes in life role expectations that increase the likelihood of mental health problems. Periods of more, or fewer, mental health problems exist in life for all people. We have used Halfon et al.’s [26] illustration, originally intended to describe changes in “health” over the life span, to visualize the life span trajectories of mental health problems and mental health (see Figure 2). Figure 2 illustrates that mental health problems can vary over time on a continuum from no problems to severe mental health problems. It is possible that neither complete wellbeing nor severe mental health problems/mental illness occur frequently. The same figure can be used to illustrate variations over the life span in mental health, defined as a state of wellbeing (see Table 1). When studying the trajectories of mental health problems in children with NDD, we are primarily interested in how mental health problems vary over time. Studying the occurrence of mental health problems may, however, not be enough—wellbeing is also important.

### 3.2. Problem 2: Distinguishing Mental Health Problems and Wellbeing as Separate Constructs in Children with NDD

We propose that mental health should not be reduced to the absence of mental illness, but should encompass variations in mental health on a continuum from low to high levels of wellbeing.

Research in positive psychology and related fields have employed numerous conceptualizations of positive mental health and wellbeing [27,28]. Each understanding of the concept may present advantages and disadvantages, and arguments for each definition could be based on validity, pragmatic aspects, logic, and so forth. One characteristic of a construct that is sometimes overlooked is whether its definition is equally valid for the full width of human experience and functioning. If the overarching goal in wellbeing research is to describe universal as well as unique aspects of human functioning, then there is relatively little utility for concepts that are only valid for a subgroup of humanity, such as typically developed adults in western countries.

#### 3.2.1. Mental Health—A Multidimensional Wellbeing Concept

The WHO’s definition of mental health [18] explicitly equates mental health with wellbeing. The WHO’s definition can be used easily when working with adults without severe cognitive impairments. It is not as easy to apply to children and adolescents within the NDD spectrum, because their ability to meet aspects of the definition—“realizing abilities, cope with stress, work productively, and make a contribution to society”—may, by definition, preclude a determination of “wellbeing”. 

Usually, wellbeing is seen as comprising positive emotional states (feeling good) [29,30,31,32] and as having fewer/lower negative emotional states [31,33]. Some authors also describe good functioning as being a part of wellbeing [29,30], including having a command over resources or achieving a balance between resources and challenges [34]. In a study of student perspectives, wellbeing was found to be related to being (e.g., happy, satisfied), having (e.g., rights, relationships, resources, voice), and doing (e.g., looking after self and others, having goals, and making good decisions) [35]. The three dimensions of being, having, and doing can apply to all people, including children with NDD, and can be linked to two dominating, broad perspectives in wellbeing research: hedonia and eudaimonia [36]. Thus, wellbeing’s positive emotional states include the two different ideas of happiness: hedonic (happiness or pleasure), that is living a pleasant life, or eudemonic (striving for, achieving something more—either personal growth or something outside the self), that is, living a goal directed or meaningful life [19]. People experience both hedonic and eudemonic happiness but may seek or value one type of wellbeing more than the other. In children and adolescents with significant NDD, the “doing” and edudaimonic elements of wellbeing may have a restricted range of expressions or require substantial support from others; however, they are not by definition excluded from the experience. 

It has been suggested that wellbeing may be best understood as a multidimensional phenomenon incorporating both ideas of wellbeing [36]. One attempt at combining hedonic and eudaimonic influences is seen in Keyes et al. [20] work. Keyes argues that mental health consists of three partly overlapping dimensions of wellbeing: emotional wellbeing (entailing positive affect, absence of negative affect, and perceived satisfaction with life), psychological wellbeing (consisting of self-acceptance, positive relations with others, personal growth, purpose in life, environmental mastery, and autonomy), and social wellbeing (social acceptance, social actualization, social contribution, social coherence, and social integration). When testing this suggestion, Keyes et al. [20] found support for a two-factor wellbeing model, corresponding to the two traditions: eudaimonia, comprising psychological and social wellbeing indicators, and hedoninia, comprising subjective (emotional) wellbeing. 

#### 3.2.2. A Dual Model of Mental Health and Mental Health Problems

Because the WHO has provided both a definition of mental disorders and a definition of mental health in which mental health is explicitly named as wellbeing, the relationship between wellbeing and mental health problems needs clarifying. Do wellbeing and mental health problems exist on the same continuum? Literature describing wellbeing as the presence of positive feelings towards your own life tends to see wellbeing as a continuum of its own. Keyes [37] considers levels of wellbeing on a scale anchored by languishing (unhappiness and experiencing difficulties) at one end and flourishing (happy and thriving—the most positive state) at the other [37,38]. Several studies provide evidence that mental health problems and wellbeing are two separate but correlated constructs, rather than one (MacArthur Foundation’s Midlife in the United States survey) [39]. Studies including children with NDD lend further support to this dual-factor model of mental health [40,41]. The term flourishing is suggested as useful to characterize people with high scores in emotional, psychological, and social wellbeing, whereas languishing can be used to categorize people with low scores on wellbeing. Thus, languishing is seen as indicating a low level of wellbeing that might, or might not, occur in conjunction with mental health problems or illness.

In conclusion, the support for the dual continua model means that we can add another layer to Figure 1. It is theoretically possible for someone to fulfill the criteria for a mental disorder (e.g., autism) and to also experience any level of mental health problems and wellbeing (circles partly overlapping). Mental illness most likely influences a person’s wellbeing, but in theory, it is possible to experience aspects of positive mental health such as wellbeing when suffering from a mental health problem. The relationship between mental health problems and the dual continua model is illustrated in Figure 3.

### 3.3. Problem 3: The Relationship between Participation and Wellbeing in Children with NDD 

We propose that participation is a key concept to relate to wellbeing because of its focus on functioning in the context of everyday activities. Participation can be an antecedent of wellbeing as well as a consequence of wellbeing.

In discussing the relationship between wellbeing and participation, we consider the antecedents and consequences of wellbeing and participation as described across diverse literature bases such as children with disabilities, aging populations, and the business literature. Antecedents and consequences provide information about possible causal links between the constructs of wellbeing and participation, although both wellbeing and participation are complexly determined and may have a cascade of effects. First, we provide descriptions of the construct of participation.

#### 3.3.1. How Participation Is Conceptualized in Various Bodies of Literature

Recent research in the child-onset disability field identifies participation as being involvement in a life situation with two dimensions. The first dimension attendance relates to the life situations and the second dimension to the involvement or engagement while being there. The dimensions are situated within the fPRC, which is neutral about the activity or life situation in which participation occurs, that is, participation can be considered in relation to any activity [5] and is pertinent for all people. 

Although participation can occur in any life situation, the need to identify the situation in which participation is being studied implies that participation is a contextually based construct. Research about participation can be found in diverse literature, for example, business literature that focuses on participation in work (e.g., [42,43], or youth delinquency research that focuses on participation in crime or the legal system [44]. Some participation research implies that participation requires others to be present—thus effectively limiting the types of life situations in which participation can be said to have occurred. For example, in the aged care/adult disability literature, there has been a focus on participation being relevant in “social”, “community”, or “complex” activities [45,46]. The fPRC describes participation as being relevant to any life situation, including activities done individually, thus providing important conceptual clarity and applicability to all people. 

Across various fields of literature are examples of studies in which the term participation is not defined explicitly: presumably based on the assumption that we all know and agree about what it is. When participation is not defined, what is measured is commonly the “attendance” dimension: that is, how often people attend particular activities, or what proportion of people attend particular situations. The notion of involvement is further explored here, because there is greater variation across literature on how involvement is operationalized compared to attendance. 

Research about participation in decision making provides one mechanism for exploring involvement. Decision making is a process—whether done collaboratively or independently—and can be relevant to any life situation. Concerning participation in community development, the implied definition of participation is both attendance (in the decision-making activity) and involvement in dialogue [47]. This perspective is consistent with youth delinquency research studies in which participation in the legal proceedings has been considered in relation to involvement in decision making and problem solving around issues directly affecting the individual [48]. A focus on collaboration in decision making is also apparent in some education literature that describes participation as children being listened to by adults and having their views considered in decision making [49]. Puritz and Majd [50] describe involvement as having a meaningful opportunity to be heard. 

Participation defined as “taking part”, which might include interacting, doing, helping, or contributing [45], or as engagement in (complex) activities [46], also provides ideas about involvement. Operationalizing these ideas, however, often results in “counting occasions of doing (something)” an idea that is closer to the notion of attending than the experience of involvement. Likewise, in the business and education literature, although the term engagement is used more commonly than participation, the focus is frequently related to “engaged time” [51]—once again a measure of attendance. In contrast, engagement defined by Russell et al. [52] as “energy in action, the connection between person and activity” (p. 1), conveys the essence of the experience of participation, and reinforces the need to consider participation in context. 

Bringing the ideas of attendance and involvement together, Bergqvist et al. [53] reported that “when a person chooses to attend an activity, it is possible for the person to be involved and that might lead to participation” (p. 1). In this example, participation is seen as a potential outcome of doing something, which suggests that participation cannot be separated from either doing or belonging. This definition of participation is consistent with the fPRC from the perspective that attendance is seen as a necessary but not sufficient condition for involvement. 

Hoogsteen and Woodgate’s [54] conceptual analysis of participation through the lens of childhood disability resulted in a definition of participation with four elements: “(i) the child must take part in something or with someone; (ii) the child must feel included or have a sense of inclusion in what they are partaking in; (iii) the child must have a choice or control over what they are taking part in; (iv) the child must work towards obtaining a personal or socially-meaningful goal or enhancing the quality of life” (pp. 329–330). The first two elements are consistent with the ideas of attendance and involvement. The third element is problematic as children often participate in activities or situations that they do not choose or control; however, the problematic nature of this element relates specifically to the attendance aspect of participation, as providing children with choices within an activity setting can help them feel involved or engaged [55,56]. This reflects an empowerment approach to the design of participation opportunities. The fourth element proposed seems closer to definitions of wellbeing than participation, but might point to the notion of future participation being driven by past and current participation—i.e., participation as a means. 

#### 3.3.2. Antecedents and Consequences to Wellbeing and Participation

The relationship between participation and wellbeing must be considered as a transactional process over time where participation at one point in time may affect wellbeing at a later point in time, and vice versa. To further consider this relationship within a process framework, two concepts that denote a causal order of events will be used: antecedents and consequences. Antecedents concern events that occur before a specified event and consequences concern events that occur after a specified event.

Antecedents to wellbeing have been described as relating to resources or contextual factors and to personal attributes. For example, having social capital and enough income [57] can support wellbeing. From a personal perspective, altruism or volunteering [32,58] and adapting to your own needs for wellbeing and your life circumstances [29] have all been identified as antecedents to wellbeing. Antecedents to low levels of wellbeing (i.e., languishing) may also be resources and contextual factors—for example, family and work variables and life stressors [32,59], limited resources [32], and lack of social engagement [60,61]. Of the factors identified as antecedents to wellbeing, participation is rarely explicitly described, although can be inferred from the literature describing altruistic behaviors, adaptive behaviors, and social engagement. Powell et al. [35] is one exception: they clearly identified participation as an antecedent contributing to wellbeing. 

Consequences of wellbeing include protection against mental health problems [62], future resilience and wellbeing [57,63], connectedness with peers [57], improved work/school productivity, engagement and achievement [59], and a sense of meaning in life [28]. Thus, one consequence of wellbeing appears to be participation; other consequences relate to personal attributes of resilience, coping, and future wellbeing.

Antecedents of participation from across the fields of literature can also be summarized as factors related to the person or the context. Person-related antecedents include interest or willingness to take part [54], and past satisfaction [64], as well as antecedents that prevent participation, such as pain [64], depression, mood disorder [32], fatigue, or physical limitations [65]. Age was proposed to shape participation in that it influences capacity for choice. Contextual antecedents of participation included initiatives that influence the physical, attitudinal, and relational environment [33,42,43]; information provided [66]; and peer modelling, family processes, socioeconomic factors, cultural practices, and governance structures [67,68]. These broad-ranging antecedents provide information about how contexts might be shaped or influenced to support participation. 

Consequences of participation as attendance included gaining skills, academic or educational achievement, health, development of self-determination or self-efficacy, overall development, and wellbeing [51,69,70,71,72]. The consequences of participation as involvement if seen as collaboration in action and decision making included impacts at the level of both person and context. For example, having agency or power and being able to contribute to choices that impact the future are personal consequences; societal transformation to realize rights and more equitable distribution of resources and benefits are contextual consequences [47,71]. Examples of consequences of participation in harmful activities were reported to include poor mental health, substance abuse, cynicism, and societal disengagement and crime [67,68]—again involving both personal and contextual consequences, strongly supporting the reciprocal nature of participation in context. 

Consequences of a lack of participation were reported to include deprivation, social injustice, limited wellbeing, lack of dignity, loss of rights [47,66], and a lack of involvement leading to lack of attendance at work or low productivity [33]. A lack of participation can lead to a lack of contribution to building social capital by particular groups in society. For example, if those with disability are not participating, their potential to shape culture, build tolerance to diversity, benefit from and contribute to common resources, and establish valued norms impacts the nature of community/society for all [73]. Additionally, the consequences of imbalanced participation, i.e., not being able to achieve balance in doing all the activities that “need” to be done and resting, included stress and mental fatigue [53]. 

#### 3.3.3. Relationships between Participation and Wellbeing

The descriptions of wellbeing are primarily focused on the person’s summative perception of their feelings about their life in terms of emotions, psychological functioning, and/or social wellbeing or a specific domain of life (e.g., recreation, work), whether focused on pleasure or striving or a combination. In contrast, descriptions of participation focus on the person taking part in context. In relation to the fPRC, wellbeing might be most closely related to ideas of “sense-of-self”, which is described as both antecedent and consequent to participation in the fPRC. The broader literature related to participation also clearly (and commonly) links wellbeing as both an antecedent (when poor [i.e., when people are languishing] it limits/reduces participation) and a consequence of participation. When participation is possible, balanced, and not in harmful activities, wellbeing (flourishing) can be enhanced. If participation is not balanced or is predominantly in harmful/negative activities, wellbeing is seen to reduce. Thus, participation and wellbeing are bi-directional: participation can influence wellbeing, and (positive) wellbeing can increase the possibility of participation [59]. 

Van Campen and Ledema [74] investigated the relationship between participation and wellbeing specifically, providing evidence about the need to understand both dimensions of participation. They focused on the impact of objective participation (attendance) on subjective wellbeing. They hypothesized a linear relationship between duration of illness leading to severity of impairment leading to objective participation leading to subjective wellbeing. Objective participation was measured as the frequency of hours in paid work, frequency of social contacts, number of holidays, and number of museum visits (thus measures of attendance). Subjective wellbeing was measured as health-related quality of life, using scales capturing mental health problems, and a measure of happiness (wellbeing). They found no empirical support for a direct relationship between objective participation and mental health problems or wellbeing. When models were adjusted to include age and socio-economic factors, a better fit was seen. In the discussion, the authors identified the need to understand subjective participation to understand its impact on wellbeing. They cited Csikszentmihalyi’s notion of flow and interpreted this finding as follows: “it is not the fact that someone participates but how they participate that determines subjective wellbeing” (p. 643). 

## 4. Implications for Measurement and Intervention with Children with NDD Following from the Three Propositions

The three problems discussed have implications for how mental health problems, wellbeing, and participation are measured in studies focusing on children with NDD. There are also implications for interventions focusing on decreasing mental health problems or enhancing wellbeing. Measurement and intervention are important topics that require consideration beyond the scope of this paper. In this section, we briefly point to some areas that need further discussion and empirical investigation.

### 4.1. Implications for Measurement: The Risk of Confusion between NDD-Core Symptoms, Mental Health Problems, and Wellbeing 

One essential aspect of any instrument aiming to measure mental health problems or screen for mental illness in children with NDD is that it should not tap into core problems associated with the NDD in question. Looking at two of the most widely used behavior problem screening questionnaires for children and adolescents, the Child Behavior Checklist (CBCL) [24] and the Strengths and Difficulties questionnaire (SDQ) [75], it is apparent that both contain several items that risk doing so (e.g., “avoids looking others in the eye” from the CBCL and ”easily distracted, concentration wanders” from the SDQ). This suggests that the problem of confusing NDD symptoms and mental health problems may apply to a substantial proportion of the research on mental health problems undertaken with children with NDDs. 

This issue is equally important when measuring wellbeing, since the presence of an NDD does not predispose individuals to either languishing or flourishing. This problem does not primarily lay within the rating scales themselves but in how data are treated. For example, concerning mental health problems, the SDQ [75] is commonly used to screen mental health problems in children with NDD, e.g., Bailey et al. [22]. In the SDQ, there are four “problem scales”: (i) hyperactivity (covering problems with both hyperactivity and inattention—the basic symptom criteria for Attention Deficit Hyperactivity Disorder), (ii) conduct problems, (iii) emotional problems (sadness, depression), and (iv) peer problems (problems in relating to peers). The subscales hyperactivity and peer problems should not be defined as mental health problems of an individual. Hyperactivity can exist along with good everyday functioning as operationalized as participation in play activities in preschool [76]. Peer problems are related to how other people react to a child and the child’s communication skills; thus, this scale is also a measure of communicative and environmental problems. For this reason, we recommend caution when drawing conclusions based on indexes, such as the internalizing or externalizing indices of the SDQ and CBCL, about mental health problems in children and adolescents with NDD diagnoses. 

### 4.2. Implications for Measurement: The Issue of Inclusiveness

A related and equally important aspect of measurement instruments is the matter of inclusiveness at the conceptual level, that is, items and scales should not preclude any level of wellbeing or mental health problems based on normative assumptions of human functioning (it should be a purely empirical question). For example, if working “productively and fruitfully” is considered by WHO [18] as a central part of wellbeing, then the individuals with the severest disabilities, for whom work in the traditional sense will never be an option, are predestined to lower levels of wellbeing. One way of reducing the risk of building conceptual barriers may be to let respondents assess wellbeing in general with a few items or using a single question. There are of course limitations to such approaches that reduce a complex phenomenon to a few items. We recommend that researchers and clinicians consider the inclusiveness of any scale chosen to measure wellbeing and mental health problems in children with NDD. Given our definition of wellbeing as subjective, we realize this recommendation is difficult to follow in the case of individuals with profound intellectual disabilities. This literature tends to use proxy-completed measures of quality of life (not wellbeing), such as the KidsLife Scale [77], which is based on a series of life domains including self-determination, social inclusion, and interpersonal relationships, in addition to material, physical, and emotional wellbeing.

Even after having considered the risk of confusing mental health problems with core symptoms of NDD and inclusiveness, the questions of inclusive measurement design and procedures remain. Many questionnaires have cognitive barriers that may make them inaccessible for children with NDDs. Instruments suited for assessing mental health problems and wellbeing in children with NDD need to be developed or adapted. In addition, manuals for how to set up structured interviews to support individuals in self-rating wellbeing and mental health problems need to be developed. One example of a questionnaire that tries to deal with these issues constructively is the Wellbeing in Special Education Questionnaire [40]. The instrument has been validated with children with mild to moderate intellectual disability and includes generic questions about wellbeing along with questions about mental health problems that could be argued to be relevant for children regardless of the level of functioning. 

Conceptual inclusiveness is also pertinent to measures of participation. Following the publication of the International Classification of Functioning, Disability, and Health [6], in which the concept participation was introduced, multiple participation measures were developed. However, the lack of conceptual clarity within the ICF led to considerable variation in approaches to measurement development [78]. One of the key issues was the conflating of the ideas of independence in performing an activity or task, with involvement in life situations (the ICF definition of participation). The problem with this approach is that children with NDD were, by definition, assessed as having poor or restricted participation simply because they were not independent (e.g., they required supports to participate due to intellectual impairment) or had limitations in their activity skills (e.g., poor manual ability). In terms of measuring participation, the inclusion of an assessment of support or aids required to participate has been critiqued in the literature [79,80]. It is considered important to conceptualize participation intrinsically and separately from other factors or variables [79]. Children with NDD may experience participation restrictions, but this should not be determined based on their skills, or attributes associated with their condition [5]. Participation attendance (being there) and involvement (the experience of participation while attending) in life situations are pertinent to all people at all phases of the life course. Measures of participation should reflect one or both these constructs.

### 4.3. Interventions Focused on Decreasing Mental Health Problems in Persons with NDD

Interventions that address mental health problems with anxiety and sadness/depression in persons with NDD are limited. Pharmacological interventions for severe mental health problems, such as antidepressant medication, may not be effective [81]. Non-pharmacological interventions have focused on talking therapies. Mindfulness (combining talking with meditation) has been shown to be effective for reducing anxiety in persons with autism [82] and a cognitive–behavioral therapy combination of on-line sessions and face to face meetings has been shown to reduce anxiety in adolescents with intellectual disability [83]. Evidence for the effect of psychotherapy is primarily limited to case-studies [84]. There is emerging evidence that talking therapies need to be modified for young people with NDD [85]. There is a dearth of evidence for talking therapies for persons with NDD who may experience more significant motor and communication difficulties. Few studies focusing on decreasing mental health problems measure wellbeing or participation of children and youth with NDD as secondary outcomes of treatment.

### 4.4. Participation Interventions as a Means to Enhancing Wellbeing in Children and Adolescents with NDD

The childhood disability literature is just beginning to explore the effects of participation interventions on wellbeing. Studies of various participation interventions, including arts-based, physical activity, life skills, coaching, and resilience-focused interventions, have provided preliminary evidence for effects on wellbeing (e.g., psychosocial well-being, self-determination, self-efficacy). For example, a scoping review of arts-based interventions for children with disabilities, which included performance (e.g., music, dance, theatre) and visual (e.g., drawing, painting, sculpting) arts-based programs, indicated that these interventions show potential to positively impact psychosocial wellbeing (i.e., emotional and social functioning), although further investigation is required with broader populations of children with physical and developmental disabilities [86]. Therapeutic horse riding, an example of a physical activity intervention, has been found to positively influence and expand the self-concepts of children with disabilities [87]. A review of the literature on therapeutic horseback riding indicates some evidence for statistically significant decreases in depression and distress, although this evidence is inconsistent and there are methodological problems in this body of research [88]. Youth with disabilities taking part in a transition-oriented life skills program have been found to have significant pre-post changes in their autonomy (as an aspect of self-determination) and self-efficacy [56]. The growing literature on coaching interventions for children and youth with disabilities focuses on engagement and goal attainment [89,90] and has yet to consider longer-term effects on wellbeing. However, the broader coaching literature indicates that participation in a cognitive–behavioral life coaching program is associated with enhanced wellbeing and quality of life [91]. Resilience-focused interventions are another promising area of intervention. A systematic review of universal resilience-focused interventions targeting child and youth wellbeing in the school setting [92] has indicated effects concerning the reduction of mental health problems.

## 5. Conclusions

This position paper suggests future directions in the scientific study of wellbeing and mental health problems in children with NDD and describes the implications for participation interventions aimed at sustaining wellbeing in children with disabilities following from the propositions:(1)Mental disorders include both diagnoses related to impairments in the developmental period, i.e., NDD and diagnoses related to mental illness. These two types of mental disorders must be separated when measuring mental health in children with disabilities. Thus, summary indexes such as externalizing and internalizing problems should be avoided, since more stable characteristics related to impairment are conflated with mental health problem indicators. Measures of mental health problems involving only mental illness indicators and not NDD impairment-related symptoms need to be developed for children diagnosed within the NDD spectrum.(2)Mental health problems and wellbeing are two related but different continua where one focuses on mental health problems and illness and the other on different degrees of wellbeing; therefore, they must be measured separately. Children with NDD, just like other people, may exhibit aspects of both mental health problems and wellbeing simultaneously. Measures of wellbeing defined as a continuum from flourishing to languishing for children with NDD need to be designed and evaluated.(3)Wellbeing and participation are distinct from each other. Wellbeing is situated within the person and can be seen as a generalized measure of a person’s mental health within generalized contexts, while participation is always situated within a more specific context or activity. The relationship between the constructs can be seen as a spiral, where participation can be both an antecedent to wellbeing and a consequence of wellbeing. Because participation is contextualized, it can be the focus of direct interventions (targeted at the context or the person) that aim to enhance wellbeing. The relationship between participation and mental health problems is hypothesized to be indirect. By increasing or sustaining participation, wellbeing can be affected. Wellbeing will lead to further participation but also act as protection from mental health problems. The proposal that participation interventions can enhance wellbeing and indirectly lessen mental health problems needs to be tested in intervention research.

## Figures and Tables

**Figure 1 ijerph-18-01656-f001:**
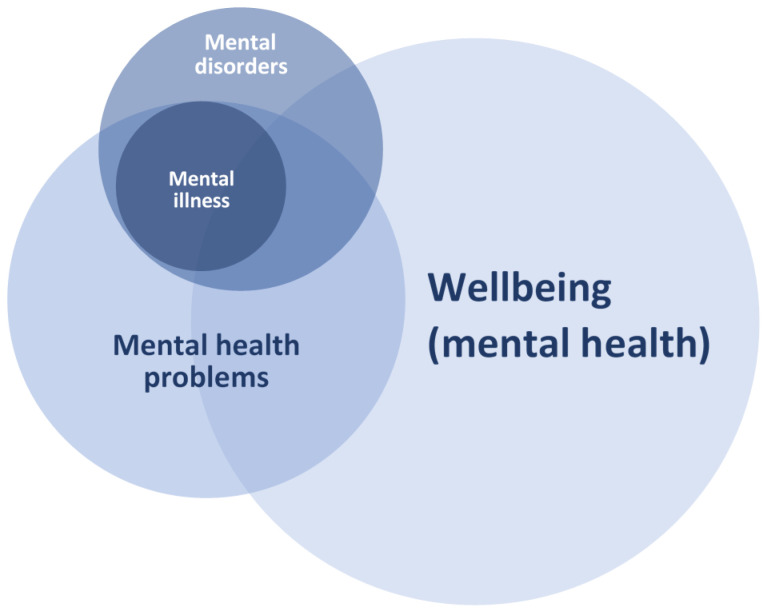
Relations between different concepts used when discussing mental health.

**Figure 2 ijerph-18-01656-f002:**
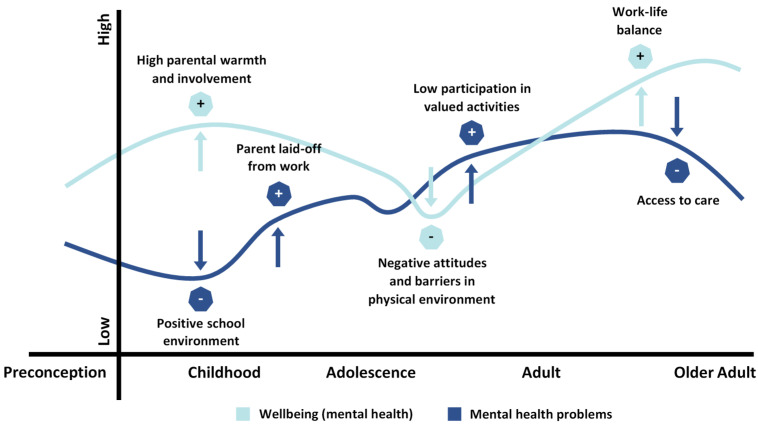
Wellbeing in a life span perspective, adapted from Halfon et al. (2014) [26].

**Figure 3 ijerph-18-01656-f003:**
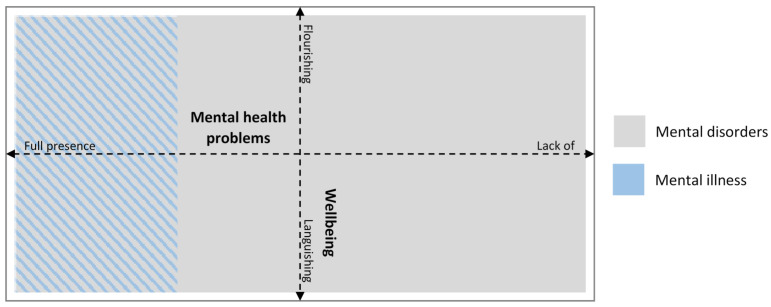
The relation of mental disorders and mental illness to the dual-continua model of wellbeing and mental health problems.

**Table 1 ijerph-18-01656-t001:** Definitions of key terms.

Mental disorder	Mental, behavioral, and neurodevelopmental disorders are syndromes characterized by clinically significant disturbance in an individual’s cognition, emotional regulation, or behavior that reflects a dysfunction in the psychological, biological, or developmental processes that underlie mental and behavioral functioning. These disturbances are usually associated with distress or impairment in personal, family, social, educational, occupational, or other important areas of functioning. ICD 11, version 09/2020, chapter 6 (http://id.who.int/icd/entity/334423054) [4]
Neurodevelopmental disorder	Neurodevelopmental disorders are behavioral and cognitive disorders that arise during the developmental period that involve significant difficulties in the acquisition and execution of specific intellectual, motor, language, or social functions. Although behavioral and cognitive deficits are present in many mental and behavioral disorders that can arise during the developmental period (e.g., Schizophrenia, Bipolar disorder), only disorders whose core features are neurodevelopmental are included in this grouping. The presumptive etiology for neurodevelopmental disorders is complex, and in many individual cases is unknown. ICD 11, version 09/2020, chapter 6 (http://id.who.int/icd/entity/1516623224) [4]
Mental illness	Mental illness (mental ill health) includes severe mental health problems and strain, impaired functioning associated with distress, symptoms and diagnosable mental disorders (e.g., schizophrenia, bipolar disorder) (European Commission, 2005) [17]
Mental health problems	A broad concept covering both less serious mental strain and more severe symptoms, fulfilling criteria for a diagnosable mental illness [9]
Mental health	“is a state of wellbeing in which an individual realizes his or her own abilities, can cope with the normal stresses of life, can work productively and fruitfully, and is able to make a contribution to his or her community’’ [18] (p. 2). Mental health defined as wellbeing vary over the life course. The description of wellbeing below is here used as an operationalization of mental health.
Wellbeing as mental health	Wellbeing’s positive emotional states include the two different ideas of happiness: hedonic (happiness or pleasure), that is living a pleasant life, or eudemonic (striving for, achieving something more—either personal growth or something outside the self), that is, living a goal directed or meaningful life [19,20].
Flourishing	“Adults with complete mental health are flourishing in life with high levels of wellbeing. To be flourishing, then, is to be filled with positive emotion and to be functioning well psychologically and socially.” [20]
Languishing	A state of a low level of wellbeing described as unhappiness and experiencing difficulties: “Adults with incomplete mental health are languishing in life with low wellbeing. Thus, languishing may be conceived of as emptiness and stagnation, constituting a life of quiet despair that parallels accounts of individuals who describe themselves and life as “hollow”, “empty”, “a shell”, and “a void” [20]. The definition focuses on low levels of wellbeing rather than expressions of mental health problems.
Participation	Involvement in a life situation comprising of two dimensions: attendance and involvement [5].
Participation as attendance	“Being there”, that is being present (physically or virtually) in the life situation [5].
Participation as involvement	The “experience of participation while attending the life situation” [5].

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
