# Peer review of "Definitions and Operationalization of Mental Health Problems, Wellbeing and Participation Constructs in Children with NDD: Distinctions and Clarifications"

_ijerph, 2021, doi:10.3390/ijerph18041656_

Round 1
Reviewer 1 Report
First of all, I would like to congratulate the authors for their work, being a pleasure to review it.
Next, I will list some comments from the review made.
Title
The main theme of the article is the study of the conceptualization of mental health and associated constructs, however it speaks at multiple points of the work of children with neurodevelopmental disorders and their health problems. The authors should address this aspect in the title of the work, since there is nothing related to children or adolescents and yet they address it in some depth throughout the article. It is essential to reformulate the title of the work so that it is a faithful reflection of the content of the manuscript.
Summary
The abstract is well established, but a last paragraph is missing that addresses the conclusions of the work, since they do not appear in the abstract and are a fundamental part of a summary of a scientific article.
Objective
The purpose of the article is to describe and define the constructs linked to mental health to ensure that future research investigating participation as a means of mental health is based on conceptual clarity. The objective of the work is adequately addressed.
Theoretical foundation
The authors make a correct theoretical foundation of the object of study and result in an order under my appropriate point of view. The reading of this section is fluent and considers that it contributes an important source of knowledge on the subject. I only wish to suggest that the authors address or add some information regarding the International Classification of Functioning, Disability and Health (ICF), since it is not referenced and consider that it should also have a small approach in this foundation.
Conclusions
The conclusions are correct but are not related to the objective or the title of the work. The authors should note that they begin their conclusions by referring to children with neurodevelopmental disorders, however this does not appear either in the title of the work or in its objective. Authors must review the entire manuscript and establish consistency between its title, objective, rationale, and conclusions.
Bibliographic references
The bibliographic style should be reviewed and DOIs should be added to the references that have it.
I suggest that the authors visit this link
https://search.crossref.org/references
and copy and paste the bibliographic references of their work there to detect which ones have DOI and incorporate them at the end of it, in order to facilitate the process of the database search engines.
Author Response
|
Reviewer comment |
Response to reviewer |
|
The main theme of the article is the study of the conceptualization of mental health and associated constructs, however it speaks at multiple points of the work of children with neurodevelopmental disorders and their health problems. The authors should address this aspect in the title of the work, The abstract is well established, but a last paragraph is missing that addresses the conclusions of the work, since they do not appear in the abstract and are a fundamental part of a summary of a scientific article. |
Good pont, title is changed
Excellent point. Coclusion has been added to abstract |
|
Objective: I only wish to suggest that the authors address or add some information regarding the International Classification of Functioning, Disability and Health (ICF), since it is not referenced and consider that it should also have a small approach in this foundation.
|
We have now introduced the ICF at the beginning of the article. We indicate how the fPRC framework is based on the ICF’s definition of participation but specifies two separate dimensions—those of attendance and involvement. |
|
The conclusions are correct but are not related to the objective or the title of the work. The authors should note that they begin their conclusions by referring to children with neurodevelopmental disorders, however this does not appear either in the title of the work or in its objective. Authors must review the entire manuscript and establish consistency between its title, objective, rationale, and conclusions.
|
Title, objective, rational and conclusions has been changed to include children with NDD |
Reviewer 2 Report
This is an interesting and thought-provoking paper on an under-researched issue. It is a non-empirical paper, discussing important conceptual issues for neurodisability research and practice. As such, it does not have results in the conventional sense, but, rather, closely argued implications. The concluded propositions are well substantiated by the preceding discussion.
The approach is highly original, in addressing a complex topic from a critical conceptual perspective. As researchers, we probably do too little in examining commonly used terms and their intersections.
The paper is a good demonstration of how discussion of theoretical concepts can generate significant messages for both research and practice. These aspects could have been developed a little further in the conclusion.
The paper is well-structured and very readable. There were two typos – see end of review.
Whilst neurodevelopmental disorders could be regarded as a minority issue, there are valuable conceptual arguments in this paper which would be applicable to disability more widely and to ageing, as well as a broader theoretical discussion on mental health, wellbeing and participation. I would, therefore, regard it as fitting within the broad scope of JERPH. I suggest, however, that the authors consider the title this paper. Not including some reference to disability or neurodisability might mean that it is not identified in database searches.
Specific comments:
Abstract: Clear and to the point
- Introduction
Given the likely readership of this journal, a brief definition or description of neurodevelopmental disorders would be useful.
The paper draws on an impressive range of research in terms of multidisciplinarity and countries of origin. This is important for a conceptual paper, in that ethnocentricity and professional silos are avoided.
Is the “problem” the paper addresses at least partly an artefact of using classification systems that as not bio-psycho-social? No, the practical example around lines 145-155 suggest not.
??considers the diagnostic relationship between ASC and some of the borderline personality disorder diagnoses??
L50-51 Both and Mindfulness, CBT and Intensive Interaction could all be regarded as non-pharmacological interventions aimed at reducing mental health problems. Sample references:
Hartley, M., Dorstyn, D. & Due, C. (2019). Mindfulness for Children and Adults with Autism Spectrum Disorder and Their Caregivers: A Meta-analysis. J Autism Dev Disord 49, 4306–4319. https://doi-org.mmu.idm.oclc.org/10.1007/s10803-019-04145-3
Nind, M. (2009). Promoting the Emotional Well-Being of People with Profound and Multiple Intellectual Disabilities: A Holistic Approach through Intensive Interaction. In J .Palwyn & S. Carnaby (eds) Profound Intellectual and Multiple Disabilities: Nursing Complex Needs. Oxford: Wiley/Blackwell. pp. 62-77.
Hronis, A., Roberts, R., Roberts, L., Kneebone, I. (2019). Fearless Me! © : A feasibility case series of cognitive behavioral therapy for adolescents with intellectual disability. Journal of Clinical Psychology, 75(6), pp. 919-932.
Mindfulness intervention is addressed at L466, but perhaps should be identified earlier.
- Classification
Interesting and significant critique of existing classification systems.
Table 1 is helpful in disaggregating and distinguishing between terms. In relation to “mental health,” how would this work for children and adults with profound intellectual disabilities, or others who may not be contributing to society is the usual, valued, ways? Addressed at least partly in line 199-204
- Core difficulties
Yes CD1: disentangling the relationship between mental disorders, and mental illness, and mental health problems is important.
3.1.1 Argued clearly and carefully. Could perhaps go further, and is clarified in Figure 1.
Figure 2 needs a little more explanation, especially in relation to NDD.
199-204 yes, this is an important point.
Being, having and doing; the latter is more complex in relation to children with significant NDD, because they are often dependent on others to support and facilitate their “doing.” Hence the quality of relationships is likely to be particularly influential. The same point would be relevant to eudaimonia.
How do Keyes’ three dimensions map onto the two factors?
L251 Given the emphasis on participation, it might be helpful for readers to reference the concept within the ICF earlier than section 4.2.
Good point regarding participation not necessarily being “social.”
L315 Interesting argument, disentangling related elements. Whilst these four elements might strengthen participation, the authors make a good argument that only the first two are essential. Element three could be regarded more as empowerment or child-directed approaches.
L332, Are the authors comfortable with an association between cognitive impairment and low wellbeing/languishing? The reference [56] refers to adults whose cognitive abilities are in decline, rather than people who have intellectual disabilities.
In the antecedents – consequences model, the authors could be a little more specific as to whether they are implying a direct contingency, or a relationship mediated by other factor/s.
L349-350 Useful suggestions for practical application.
L361 Are there consequences for society/communities if certain (groups of) individuals are not able to participate or choose not to participate?
- Implications
Section 4.1 is particularly important for diagnostic, assessment and intervention work with children with NDD.
L436-> This is indeed a significant issue in relation to children with profound intellectual disabilities, where direct self report may not be feasible. The authors could consider QoL measures designed specifically for this group (Petry, Maes & Vlaskamp, 2009; Gomez et al, 2016), though this does raise the question of the relationship between QoL and wellbeing.
L454 Again, this is a very significant issue. The authors might consider the possibility of parallel measure that indicate the level of support required by a child in order to achieve participation in specific activities.
L474 Would creative arts therapies be relevant here? Admittedly, they are poorly evaluated.
- Conclusions
The propositions are well supported by the arguments in the paper. Perhaps the authors could include some of the sound points regarding implications in earlier sections.
Typos
L411 Lay->lie
L550 Grasnlund -> Granlund
Author Response
Thank you for your very constructive feedback. Please see attachment

Reviewer 3 Report
First of all, I want to thank the authors for their topic treatment, because the results provide an advance in clarification current knowledge about mental health problems, wellbeing, participation and Child with neurodevelopmental disorders.
The purpose of this paper was to describe and define the conceptual relations between mental health problems, mental health (defined as wellbeing), and participation to ensure that future research investigating participation as a means to mental health is founded on conceptual clarity.
This paper suggests future directions in the scientific study of wellbeing and mental health problems in children with neurodevelopmental disorders. Thus the implications for participation interventions were described in order to aimed at sustaining wellbeing in children with disabilities from the following propositions:
Firstly, it was shown that mental disorders include both diagnoses related to impairments in the developmental period and diagnoses related to mental illness. The evidence show that these two types of mental disorders must be separated when measuring mental health in children with disabilities.
Secondly, it was shown that mental health problems and wellbeing are two related but different continua between them. Thus they must be measured separately, because one of them focuses on mental health problems and illness and the other one on different degrees of wellbeing.
Finally, wellbeing and participation are related but they are distinct from each other. Wellbeing is situated within the person and participation is always situated within a more specific context or activity. The relationship between both concepts is that participation can be both an antecedent to wellbeing and a consequence of wellbeing such as by increasing or sustaining participation, wellbeing can be affected.
The study (a concept paper) is correctly designed and technically sound. The question is original and well defined and the article purpose provide an advance in current knowledge.
The article was written in an appropriate way and the results were presented appropriately and the conclusions will be interesting for the readership of the Journal and it will attract a wide readership related with this topic.
Therefore, it is considered that there will be a general benefit with the publication of this article.
Author Response
Thank you for your very positive comments to this manuscript. We found no comments that we needed to respond to
Round 2
Reviewer 1 Report
Dear authors,
Once the manuscript and the changes made by the authors have been reviewed, I believe that the article can be published if the editor and the other reviewers consider it.
Congratulations to the authors
Important, do not forget to make the changes in the metadata of the journal platform, in relation to the abstract.